# Counting the Optimal Solutions in Graphical Models

**Radu Marinescu**
IBM Research
Dublin, Ireland
radu.marinescu@ie.ibm.com

**Rina Dechter**
University of California, Irvine
Irvine, CA 92697, USA
dechter@ics.uci.edu

## Abstract

We introduce #opt, a new inference task for graphical models which calls for counting the number of optimal solutions of the model. We describe a novel variable elimination based approach for solving this task, as well as a depth-first branch and bound algorithm that traverses the AND/OR search space of the model. The key feature of the proposed algorithms is that their complexity is exponential in the induced width of the model only. It does not depend on the actual number of optimal solutions. Our empirical evaluation on various benchmarks demonstrates the effectiveness of the proposed algorithms compared with existing depth-first and best-first search based approaches that enumerate explicitly the optimal solutions.

## 1   Introduction

Graphical models such as belief networks, Markov networks, constraint networks or influence diagrams provide a powerful framework for reasoning with probabilistic and deterministic information. Combinatorial optimization tasks such as finding the minimum or maximum cost solutions arise in many applications and often can be efficiently solved by search or variable elimination schemes.

Although finding the optimal solution is paramount in many practical situations, we argue that it is important to also know how many optimal solutions there are and, possibly to enumerate all or just a fraction thereof. Indeed, in genetic linkage analysis finding the number of maximum likelihood haplotype configurations may shed additional light on how the genetic information is transmitted from ancestors to descendants in a pedigree [1]. In computational protein design finding the number of optimal protein side-chain resonance assignments could be indicative for the protein structure determination [2]. Similarly, knowing the number of optimal frequency assignments to the radio links in a communication network could help the engineers produce more reliable network designs [3]. In post-optimality analysis we may be interested in estimating the distribution of optimal solutions over the values of a certain target variable, for which we clearly require the count of optimal solutions. The number of optimal solutions may also be used as a feature for explaining the hardness of finding an optimal solution to a problem instance. It thus may be employed to guide a random problem generator to produce hard problem instances for optimization.

One approach that gained attention in the past decade has focused on knowledge compilation techniques that produce a more compact representation of all optimal solutions [4, 5]. In particular, [5] described an efficient way to compile a graphical model into a compact AND/OR Multi-Valued Decision Diagram (AOMDD) which represents all its optimal solutions. More recently, [6] introduced a collection of depth-first and best-first search algorithms for computing the set of $m$-best solutions of a graphical model. However, both compilation based approaches or specialized $m$-best algorithms will yield a count of the optimal solutions by enumeration. In particular, the compilation based techniques typically count the number of optimal solutions during a secondary pass over the compiled decision diagram, whereas the $m$-best algorithms must rely on a sequence of searches each with a different value of $m$ in order to recover the actual number of optimal solutions. In contrast, the algorithms we will present and explore are not dependent on the number of optimal solutions.

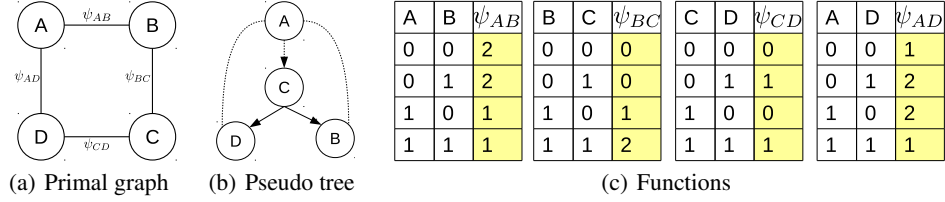

| A | B | $\psi_{AB}$ |   | B | C | $\psi_{BC}$ |   | C | D | $\psi_{CD}$ |   | A | D | $\psi_{AD}$ |
|---|---|---|---|---|---|---|---|---|---|---|---|---|---|---|
| 0 | 0 | 2 |   | 0 | 0 | 0 |   | 0 | 0 | 0 |   | 0 | 0 | 1 |
| 0 | 1 | 2 |   | 0 | 1 | 0 |   | 0 | 1 | 1 |   | 0 | 1 | 2 |
| 1 | 0 | 1 |   | 1 | 0 | 1 |   | 1 | 0 | 0 |   | 1 | 0 | 2 |
| 1 | 1 | 1 |   | 1 | 1 | 2 |   | 1 | 1 | 1 |   | 1 | 1 | 1 |

(a) Primal graph     (b) Pseudo tree                 (c) Functions

Figure 1: A simple graphical model.

**Contributions** In this paper, we define the `#opt` task for graphical models, and show how the common algorithmic principles of variable elimination and search can be extended to this task. Specifically, the idea behind variable elimination for `#opt` is to first capture all the optimal solutions as a constraint representation by flattening the relation of the cost-to-go functions and then apply the counting variable elimination algorithm over the resulting set of constraints. In our algorithm these two phases are interleaved. Subsequently, we present a depth-first branch and bound algorithm that traverses an AND/OR search space of the graphical model. For each node the algorithm computes a pair of values, one representing the optimal cost below the node in the search space and the other the number of the corresponding optimal solutions. We also provide a formulation of the `#opt` task within the semiring framework [7] thus placing it in relation to other well known graphical models tasks. It is well known that counting the number of *all* solutions in a graphical model can be done without enumeration in complexity exponential in the induced width of the model [8, 9]. We show here that counting the number of *optimal* solutions can be done without enumeration as well, with the same complexity. An empirical evaluation on various benchmark problems demonstrates the effectiveness of the proposed algorithms compared with enumeration-based extensions of depth-first and best-first search to this task.

The `#opt` task involves both summation and optimization. Other tasks involving these two operations are Marginal MAP and finding optimal policies that maximize the expected utility, defined over influence diagrams [10, 11, 12]. But, there is a notable difference. While, as we will show, `#opt` can be formulated within the semiring framework [7] and solved by traditional algorithms exactly [13], it does not land itself to the common approximation schemes such as the mini-bucket approach [14], at least not in a straightforward manner. This, we think makes it more unique and thus deserving a more focused attention.

## 2 Background

A *graphical model* is a tuple $\mathcal{M} = \langle \mathbf{X}, \mathbf{D}, \mathbf{F} \rangle$, where $\mathbf{X} = \{X_i : i \in V\}$ is a set of variables indexed by set $V$ and $\mathbf{D} = \{D_i : i \in V\}$ is the set of their finite domains of values. $\mathbf{F} = \{\psi_\alpha : \alpha \in F\}$ is a set of discrete real-valued local functions defined on subsets of variables, where we use $\alpha \subseteq V$ and $\mathbf{X}_\alpha \subseteq \mathbf{X}$ to indicate the *scope* of function $\psi_\alpha$, i.e., $\mathbf{X}_\alpha = var(\psi_\alpha) = \{X_i : i \in \alpha\}$. A *solution* is a complete assignment to the variables, namely $\mathbf{x} = (X_1 = x_1, \ldots, X_n = x_n)$. Given a set of variables $\mathbf{S} = \{X_1, \ldots, X_k\}$, we denote by $\Omega(\mathbf{S})$ the Cartesian product of their domains, namely $\Omega(\mathbf{S}) = D_1 \times \cdots \times D_k$. The function scopes yield a *primal graph* $G$ whose vertices are the variables and whose edges connect any two variables that appear in the scope of the same function. The graphical model $\mathcal{M}$ represents a *global function*, whose scope is $\mathbf{X}$ and which is the combination of all the local functions, namely: $\mathcal{F}(\mathbf{x}) = \sum_{\alpha \in F} \psi_\alpha(\mathbf{x}_\alpha)$.

The most common optimization task (`opt`) for graphical models is to compute the *optimal value* $V^* = \min_\mathbf{x} \mathcal{F}(\mathbf{x})$ and its optimizing configuration $\mathbf{x}^* = \operatorname{argmin}_\mathbf{x} \mathcal{F}(\mathbf{x})$. The latter is also known as the *optimal solution*.

However, a graphical model may have more than one optimal solution. We next define formally the task of counting the number of optimal solutions of a graphical model which we abbreviate hereafter by `#opt`.

DEFINITION 1 (`#opt`). *Given a graphical model $\mathcal{M} = \langle \mathbf{X}, \mathbf{D}, \mathbf{F} \rangle$, the `#opt` task is to compute $|\mathcal{S}|$, where $\mathcal{S} = \{\mathbf{x} \mid \mathcal{F}(\mathbf{x}) = V^*, V^* = \min_\mathbf{x} \sum_{\alpha \in F} \psi_\alpha(\mathbf{x}_\alpha)\}$.*

We also define the task `e-opt` which calls for enumerating explicitly all the optimal solutions:

DEFINITION 2 (e-opt). *Given a graphical model* $\mathcal{M} = \langle \mathbf{X}, \mathbf{D}, \mathbf{F} \rangle$, *the* e-opt *task is to enumerate all elements in the set* $\mathcal{S} = \{\mathbf{x} \mid \mathcal{F}(\mathbf{x}) = V^*, V^* = \min_{\mathbf{x}} \sum_{\alpha \in F} \psi_\alpha(\mathbf{x}_\alpha)\}$.

An important feature of graphical models in characterizing complexity is the *induced width* (or *treewidth*). The induced graph of $G$ relative to an ordering $\tau$ of its variables is obtained by processing the nodes in reverse order of $\tau$. For each node all its earlier neighbors are connected, including neighbors connected by previously added edges. The width of a node is the number of edges connecting it to nodes lower in the ordering. The *induced width* of $G$ along $\tau$, denoted $w^*(\tau)$, is the maximum width of the nodes in the induced graph.

**Example 1.** *Figure 1(a) depicts the primal graph of a simple graphical model representing a global function over 4 variables* $\mathbf{X} = \{A, B, C, D\}$ *with 4 local functions (shown in Figure 1(b)) defined by the arcs (each pair is a scope of one function). There are 6 optimal solutions with optimal value 3, namely* $\mathcal{S} = \{0000, 0010, 1000, 1001, 1010, 1011\}$.

We describe next brute-force search and variable elimination based schemes for counting the optimal solutions.

**Depth-First Branch and Bound** Solving #opt can be done by a simple extension of depth-first branch and bound search [15]. The algorithm, called BnB, traverses the space of partial assignments in a depth-first manner while maintaining an upper bound $U$ on the optimal value and a counter $c$ which is updated every time a new solution value $V$ is found: $c = c + 1$ if $V = U$ and $c = 1$ if $V < U$, respectively. In the latter case, $V$ becomes the current best upper bound. Throughout the search, the algorithm also attempts to prune unpromising regions of the search space. Namely, at each node $n$ it computes a heuristic lower bound $f(n)$ of the best solution extending the current partial assignment and prunes the respective subtree if the heuristic estimate is strictly greater than the current upper bound ($f(n) > U$). The strict inequality is required in order to find all optimal solutions. When search terminates, the value of the counter $c$ gives the number of optimal solutions.

**Best-first Search** Alternatively, we can use a best-first search strategy such as A* search [16]. Algorithm A* for #opt maintains the search frontier in the OPEN list and always expands first the node $n$ with the smallest f-value $f(n) = g(n) + h(n)$, where $g(n)$ is the cost of the path from the root of the search space to $n$ and $h(n)$ is a lower bound of the best extension to a solution [17]. When the optimal solution is encountered, the algorithm saves its value $V^*$, initializes a counter $c$ to 1 and continues the search. Every time a new optimal solution is found, the counter $c$ is incremented. Search terminates when OPEN is empty, in which case $c$ gives the number of optimal solutions.

**Variable Elimination** An immediate extension to #opt within the compilation paradigm is to apply a variable elimination scheme for optimization along an ordering $\tau$ and then enumerate all optimal solutions one by one in a greedy fashion using the intermediate messages generated during the elimination procedure [13]. The only difference between this algorithm and a regular variable elimination is that the forward decoding pass does not terminate with the first optimal solution.

**Complexity of Brute-Force Approaches** All the algorithms presented above require enumerating all the optimal solutions and thus have a factor of $\#opt$ in their complexity. If $|T|$ captures the search space size explored by best-first search for finding the first optimal solution and $\#opt$ is their number, best-first can be bounded by $O(|T| + \#opt)$. Branch and bound is often faster and has a better memory management than best-first search yet it is harder to bound and we cannot give a better bound than $O(|T| \cdot \#opt)$. The simple variable elimination algorithm we described generates a compiled representation in time and memory exponential in the induced width $w_\tau^*$ along an ordering $\tau$. Then, generating each new optimal solution requires consulting the compiled structure and is in the worst-case exponential in $w_\tau^*$ implying an overall worst-case complexity of $O(n \cdot k^{w_\tau^*} \cdot \#opt)$, where $n$ is the number of variables and $k$ bounds the domain size.

## 3 Bucket Elimination for #opt

The main drawback of the brute-force search and variable-elimination approaches described in Section 2 is that they must enumerate explicitly all optimal solutions (i.e., they actually solve the e-opt task). If the number of optimal solutions is large then the computational overhead can be significant. In this and the next section we will describe more efficient methods for solving #opt that are based on either bucket elimination or on depth-first branch and bound search over an AND/OR search spaces, but avoid explicit enumeration of the optimal solutions.

---

**Algorithm 1** BE for `#opt`

**Require:** graphical model $\mathcal{M} = \langle \mathbf{X}, \mathbf{D}, \mathbf{F} \rangle$, elimination order $\tau = X_1, \ldots, X_n$
 1: Let $\Psi = \{\psi_\alpha : \psi_\alpha \in \mathbf{F}\}$
 2: **for all** variable $X_p$ in the reversed order $\tau$ **do**
 3:     Create bucket $\mathcal{B}_p$ and its associated set $\Psi_p$
 4:     Let $\Psi_p = \{\psi_\alpha \in \Psi : X_p \in vars(\psi_\alpha)\}$
 5:     Let $\Psi \leftarrow \Psi \setminus \Psi_p$, $\Lambda_p = \emptyset$ and $\Gamma_p = \emptyset$
 6: **for all** variable $X_p$ in the reversed order $\tau$ **do**
 7:     Let $\Psi_p = \{\psi_1 \ldots \psi_r\}$, $\Lambda_p = \{\lambda_1 \ldots \lambda_m\}$ and $\Gamma_p = \{\gamma_1, \ldots, \gamma_q\}$
 8:     Let $\psi_p \leftarrow \sum_{i=1}^{r} \psi_i$
 9:     Let $\lambda_p \leftarrow \min_{x_p}(\psi_p + \sum_{j=1}^{m} \lambda_j)$
10:     Let $\gamma_p \leftarrow \sum_{x'_p}(\bar{\psi}_p \cdot \prod_{k=1}^{q} \gamma_k)$ where $x'_p \in \operatorname{argmin}_{x_p}(\psi_p + \sum_{i=1}^{r} \psi_i)$ and $\bar{\psi}_p$ is the flattened $\psi_p$ (see text below for details)
11:     Add $\lambda_p$ and $\gamma_p$ to the sets $\Lambda$ and $\Gamma$ of the highest bucket corresp. to a variable in $vars(\lambda_p)$; If $X_p$ is the first variable then add $\lambda_p$ to $\Lambda_0$ and $\gamma_p$ to $\Gamma_0$
12: Let $v^* \leftarrow \sum_{\lambda \in \Lambda_0} \lambda$ and $c^* \leftarrow \prod_{\gamma \in \Gamma_0} \gamma$
13: **return** $\langle v^*, c^* \rangle$

---

As noted, the optimal solution to a graphical model can be obtained by using the *bucket elimination* (BE) algorithm which eliminates (minimizes over) the variables in sequence [13]. In order to extend it to counting the optimal solutions and avoid enumeration we use the concept of *flattened* representation of a function. Specifically, given a function $\psi$, its flattening denoted by $\bar{\psi}$ is defined as: $\bar{\psi}(\mathbf{y}) = 1$ if $\psi(\mathbf{y}) \neq \infty$ and $\bar{\psi}(\mathbf{y}) = 0$ otherwise, for all $\mathbf{y} \in \Omega(vars(\psi))$. When a variable $X$ is eliminated, we first record a standard cost message that represents the cost-to-go corresponding to minimizing over the domain values of $X$. Then, we count the minimizing configurations of $X$ and record them in a new count message which has the same scope as the cost message. The latter can be viewed as a counting step over a constraint representation of the flattened cost-to-go function.

Algorithm 1 presents the BE procedure for solving `#opt`. Given a variable ordering $\tau = X_1, \ldots X_n$, the functions are first partitioned into their corresponding buckets such that a bucket $\mathcal{B}_p$ is associated with a single variable $X_p$ and a function is placed in the bucket of its argument that appears latest in the ordering (lines 1–5). With each bucket $\mathcal{B}_p$ we also assign three sets $\Psi_p$, $\Lambda_p$ and $\Gamma_p$ to store the original functions as well as the cost and count messages, respectively.

The algorithm processes each bucket in reversed order, from last to first by a variable elimination procedure that computes a new *cost* message and a new *count* message which are both placed into a lower bucket. Specifically, let $X_p$ be the current variable and let $\psi_p$ be the bucket function which is obtained by summing all original functions in the bucket (line 8). The *cost* message (or $\lambda$-message) $\lambda_p$ computed in bucket $\mathcal{B}_p$ is obtained by minimizing out variable $X_p$ from the compound function that combines by summation the bucket function and all incoming cost messages to this bucket. Namely, $\lambda_p \leftarrow \min_{x_p}(\psi_p + \sum_{j=1}^{m} \lambda_j)$. This is the usual cost-to-go message when computing the optimal solution. The *count* message (or $\gamma$-message) originating from $\mathcal{B}_p$ is computed by first multiplying all the incoming count messages to this bucket with the flattened bucket function $\bar{\psi}_p$ and then summing over the bucket's variable. Namely, we have $\gamma_p \leftarrow \sum_{x'_p}(\bar{\psi}_p \cdot \prod_{k=1}^{q} \gamma_k)$. Notice that the summation is performed only over the minimizing domain values $x'_p$ of $X_p$, namely over $x'_p \in \operatorname{argmin}_{x_p}(\psi_p + \sum_{i=1}^{r} \psi_i)$. Finally, after the first variable in the ordering is processed, the sets $\Lambda_0$ and $\Gamma_0$ contain all the constant messages generated during the execution. Therefore, the optimal value is obtained by summing up all constants in $\Lambda_0$ (line 12), while the number of optimal solutions is calculated as the product of all constants in $\Gamma_0$ (line 12), respectively.

We can show that BE for `#opt` is time and space exponential in the induced width $w_\tau^*$ of the ordering $\tau$, i.e., $O(n \cdot k^{w_\tau^*})$, where $n$ is the number of variables and $k$ bounds the domain size.

## 4  AND/OR Branch and Bound for `#opt`

Significant recent improvements in search for optimization in graphical models have been achieved by using AND/OR search spaces, which often capture problem structure far better than standard OR search methods [18, 19, 20, 21]. We next introduce a depth-first branch and bound algorithm that traverses an AND/OR search space to compute the number of optimal solutions as well as the optimal value.

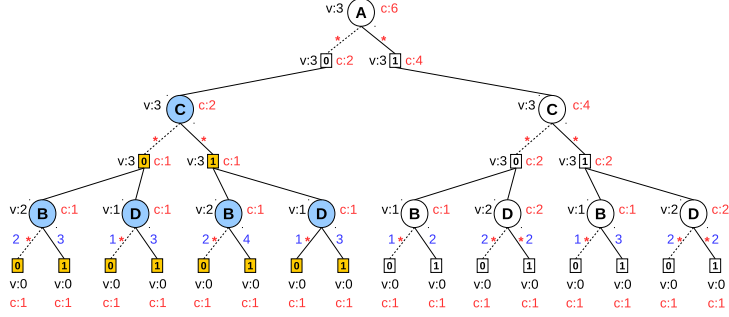

Figure 2: AND/OR search tree of the problem from Figure 1.

**AND/OR Search Spaces** The AND/OR search space is defined relative to a *pseudo tree* of the primal graph, which captures problem decomposition.

DEFINITION **3** (pseudo tree). *A pseudo tree of an undirected graph $G = (V, E)$ is a directed rooted tree $\mathcal{T} = (V, E')$ such that every arc of $G$ not included in $E'$ is a back-arc in $\mathcal{T}$ connecting a node in $\mathcal{T}$ to one of its ancestors. The arcs in $E'$ may not all be included in $E$.*

Given a graphical model $\mathcal{M} = \langle \mathbf{X}, \mathbf{D}, \mathbf{F} \rangle$ with primal graph $G$ and pseudo tree $\mathcal{T}$ of $G$, the *AND/OR search tree* $S_{\mathcal{T}}$ based on $\mathcal{T}$ has alternating levels of OR nodes corresponding to the variables, and AND nodes corresponding to the values of the OR parent's variable, with arc weights extracted from the original functions $\mathbf{F}$. Identical subproblems, identified by their *context* (the partial instantiation that separates the subproblem from the rest of the problem graph), can be merged, yielding an *AND/OR search graph*. Merging all context-mergeable nodes yields the *context minimal AND/OR search graph*, denoted $C_{\mathcal{T}}$. The size of $C_{\mathcal{T}}$ is exponential in the induced width of $G$ along a depth-first traversal of $\mathcal{T}$ (see also [18]).

A solution tree $\hat{x}$ is a subtree of $C_{\mathcal{T}}$ such that: (1) contains the root of $C_{\mathcal{T}}$; (2) if an internal OR node $n \in C_{\mathcal{T}}$ is in $\hat{x}$ then exactly one of its AND children is in $\hat{x}$; (3) if an internal AND node $n \in C_{\mathcal{T}}$ is in $\hat{x}$ then all its OR children are in $\hat{x}$; (4) every tip node in $\hat{x}$ is a terminal node.

### 4.1 Arc Weights and Node Values

The OR-to-AND arcs in the AND/OR search space are associated with *weights* that are defined based on the graphical model's functions [19]. In order to solve the #opt task, each node $n$ in $C_{\mathcal{T}}$ is associated with two *values* denoted by $v(n)$ and $c(n)$, respectively. The optimal value below $n$ is given by $v(n)$, while $c(n)$ captures the number of optimal solutions of the conditioned subproblem rooted at $n$. Based on previous work [18], the value $v(n)$ can be computed recursively based on the values of $n$'s successors in $C_{\mathcal{T}}$, as follows:

$$v(n) = \begin{cases} 0, \text{if } n \text{ is terminal AND node} \\ \sum_{m \in succ(n)} v(m), \text{if } n \text{ is AND node} \\ \min_{m \in succ(n)} (w_{(n,m)} + v(m)), \text{if } n \text{ is OR node} \end{cases} \tag{1}$$

Similarly, we can compute $c(n)$ recursively as:

$$c(n) = \begin{cases} 1, \text{if } n \text{ is terminal AND node} \\ \prod_{m \in succ(n)} c(m), \text{if } n \text{ is AND node} \\ \sum_{m' \in succ(n)} c(m), \text{if } n \text{ is OR node, } and \text{ m'} \in \operatorname{argmin}_{m \in succ(n)} (w_{(n,m)} + v(m)) \end{cases} \tag{2}$$

Clearly, the values $v(s)$ and $c(s)$ of the root node $s$ represent the optimal value and the number of optimal solutions of the initial problem.

**Example 2.** *Figure 2 displays the weighted AND/OR search tree of the problem from Figure 1 based on the pseudo tree from Figure 1(b). The node values $v(n)$ and $c(n)$ are shown next to each node (in black and red, respectively). Consider the highlighted subtree rooted at the OR node labeled $C$. The optimal value of the subproblems rooted by its AND children is 3, and since both values of $C$ are optimal in this case it follows that the number of optimal solutions below $C$ is 2.*

---

**Algorithm 2** AOBB for `#opt`

---

**Require:** graphical model $\mathcal{M} = \langle \mathbf{X}, \mathbf{D}, \mathbf{F} \rangle$, pseudo tree $\mathcal{T}$, heuristic $h(n)$
1: **function** AOBB($\hat{x}, \mathbf{X}, \mathbf{D}, \mathbf{F}$)
2:　**if** $\mathbf{X} = \emptyset$ **then**
3:　　**return** $(0, 1)$
4:　**else**
5:　　$X_i \leftarrow SelectVar(\mathcal{T})$
6:　　Let $n$ be the OR node labeled by $\langle X_i \rangle$
7:　　**if** $Ctxt(n)$ in cache **then**
8:　　　$(v(n), c(n)) \leftarrow ReadCache(Ctxt(n))$
9:　　**else**
10:　　　$v(n) \leftarrow \infty; c(n) \leftarrow 0; ch(n) \leftarrow \emptyset$
11:　　　**for all** domain value $x_i \in D_i$ **do**
12:　　　　Extend partial solution $\hat{x} \leftarrow \hat{x} \cup \{x_i\}$
13:　　　　Let $m$ be the AND node $\langle X_i, x_i \rangle$
14:　　　　Let $v(m) \leftarrow w_{(n,m)}; c(m) \leftarrow 1$
15:　　　　Calculate $f(\hat{x})$ using the $h(m)$ of the unexpanded leaves $m$ of $\hat{x}$
16:　　　　**if** $f(\hat{x}) \leq v(s)$ **then**
17:　　　　　**for all** $k = 1 \ldots q$ **do**
18:　　　　　　$(v, c) \leftarrow AOBB(\hat{x}, \mathbf{X}_k, \mathbf{D}_k, \mathbf{F}_k)$
19:　　　　　　$v(m) \leftarrow v(m) + v$
20:　　　　　　$c(m) \leftarrow c(m) \times c$
21:　　　　　　$ch(n) \leftarrow ch(n) \cup \{m\}$
22:　　　$v(n) \leftarrow \min_{m \in ch(n)} v(m)$
23:　　　$c(n) \leftarrow \sum_{m' \in ch(n)} c(m')$, where $m' \in \operatorname{argmin}_{m \in ch(n)} v(m)$
24:　　$WriteCache(Ctxt(n), v(n), c(n))$
25:　　**return** $(v(n), c(n))$

---

**Branch and Bound Search** The depth-first AND/OR branch and bound (AOBB) search method for the `#opt` task is described by Algorithm 2. The following notation is used: $(\mathbf{X}, \mathbf{D}, \mathbf{F})$ is the problem with which the procedure is called, $\hat{x}$ is the current partial solution subtree, $Ctxt(n)$ denotes the context of a node $n$, while $v(n)$ and $c(n)$ are the node values that are updated based on the values of their successors in the search space (see also Equations 1 and 2). The weight $w_{(n,m)}$ labels the arc from the OR node $n$ to its AND child $m$. The algorithm assumes that variables are selected according to a pseudo tree $\mathcal{T}$. If the set $\mathbf{X}$ is empty, then the result is trivially computed (lines 2–3). Else, AOBB selects a variable $X_i$ and expands the OR node $n$ labeled by $X_i$, namely it iterates over its domain values $x_i$ (line 11) to compute the node values $v(n)$ and $c(n)$, respectively. The algorithm attempts to retrieve the results cached at the OR nodes (lines 7–8). If a valid cache entry is found for the current OR node then the node values are updated (line 8) and the search continues. Before expanding the AND node $m$ labeled by $\langle X_i, x_i \rangle$, AOBB uses the $h(\cdot)$ values of the unexpanded leaf nodes in $\hat{x}$ to compute the heuristic evaluation function $f(\hat{x})$ which yields a lower bound on the optimal extension of $\hat{x}$. Subsequently, it safely prunes the search space below $m$ if $f(\hat{x}) > v(s)$, where $v(s)$ is the current value of the root node $s$ and is an upper bound on the optimal solution value. Notice that, unlike in regular branch and bound search, a strict inequality is required to account for all optimal solutions. The problem is then decomposed into a set of $q$ independent subproblems, one for each child $X_k$ of $X_i$ in the pseudo tree, which are then solved sequentially (line 17). After trying all feasible values of variable $X_i$, the minimal cost as well as the number of optimal solutions to the problem rooted by $X_i$ remain in $v(n)$ and $c(n)$, which are returned (line 25). Based on previous work [18], we can show that the complexity of AOBB is time and space $O(n \cdot k^{w^*_\mathcal{T}})$, where $n$ is the number of variables, $k$ bounds the domain size and $w^*_\mathcal{T}$ is the induced width along a depth-first traversal of the pseudo tree.

Although algorithms AOBB and BE have the same worst-case complexity for `#opt`, in practice, AOBB is likely to be more effective than BE because it can exploit a heuristic evaluation function to prune the search space. We will illustrate this experimentally on several problem benchmarks.

## 5  The Semiring Formulation for `#opt`

We next show how to formulate and solve the `#opt` task within a semiring based system [7]. Specifically, consider the semiring $\mathcal{A} = \langle \mathbb{R}^2, \otimes, \oplus \rangle$ over pairs of real values, where operations $\otimes$ and $\oplus$ are defined as follows (intuitively, $v$ is the cost of a solution and $c$ is the count):

$$(v_1, c_1) \otimes (v_2, c_2) = (v_1 + v_2, c_1 \cdot c_2) \quad (3) \quad (v_1, c_1) \oplus (v_2, c_2) = \begin{cases} (v_1, c_1 + c_2), & \text{if } v_1 = v_2 \\ (v_1, c_1), & \text{if } v_1 < v_2 \\ (v_2, c_2), & \text{if } v_1 > v_2 \end{cases}$$

$$(4)$$

It is easy to verify that $\otimes$ and $\oplus$ are commutative and associative, namely $a \otimes b = b \otimes a$, $a \oplus b = b \oplus a$, $a \otimes (b \otimes c) = (a \otimes b) \otimes c$ and $a \oplus (b \oplus c) = (a \oplus b) \oplus c$, for all $a, b, c \in \mathbb{R}^2$, respectively. In order to facilitate local computations over the semiring valuations, $\otimes$ must distribute over $\oplus$ [22].

PROPOSITION 1 (distributivity). *Given a semiring $\mathcal{A} = \langle \mathbb{R}^2, \otimes, \oplus \rangle$, then $\forall a, b, c \in \mathbb{R}^2$ we have that $a \otimes (b \oplus c) = (a \otimes b) \oplus (a \otimes c)$.*

*Proof.* Let $a = (v_1, c_1)$, $b = (v_2, c_2)$ and $c = (v_3, c_3)$. Assume that $v_2 < v_3$ (the other cases $v_2 > v_3$ and $v_2 = v_3$ can be shown in a similar manner). Clearly, $v_1 + v_2 < v_1 + v_3$ also holds. Then $a \otimes (b \oplus c) = (v_1, c_1) \otimes (v_2, c_2) = (v_1 + v_2, c_1 \cdot c_2)$. We have that $(a \otimes b) \oplus (a \otimes c) = (v_1 + v_2, c_1 c_2) \oplus (v_1 + v_3, c_1 \cdot c_3) = (v_1 + v_2, c_1 \cdot c_2)$, which concludes the proof. $\qquad\square$

Given a graphical model $\langle \mathbf{X}, \mathbf{D}, \mathbf{F} \rangle$, each local function $\psi_\alpha(\mathbf{X}_\alpha) \in \mathbf{F}$ can be expressed as a semiring valuation $\phi_\alpha : \Omega(\mathbf{X}_\alpha) \to \mathbb{R}^2$ such that $\forall \mathbf{y} \in \Omega(\mathbf{X}_\alpha)$, $\phi_\alpha(\mathbf{y}) = (\psi_\alpha(\mathbf{y}), 1)$. We use the two operations $\oplus$ and $\otimes$ in $\mathcal{A}$ to define the combination and elimination operators, as follows. Given two valuations $\phi_1(\mathbf{Y})$ and $\phi_2(\mathbf{Z})$ such that $\mathbf{Y}, \mathbf{Z} \subseteq \mathbf{X}$, the *combination* $\bigotimes$ is defined by $\phi_1 \bigotimes \phi_2(\mathbf{yz}) = \phi_1(\mathbf{y}) \otimes \phi_2(\mathbf{z})$ for all $\mathbf{y} \in \Omega(\mathbf{Y})$ and $\mathbf{z} \in \Omega(\mathbf{Z})$. Similarly, given a valuation $\phi(\mathbf{Y})$ such that $\mathbf{Y} \subseteq \mathbf{X}$ and $\mathbf{Y} = \{X\} \cup \mathbf{Z}$, the *elimination* $\bigoplus$ is defined by $\bigoplus_X \phi(\mathbf{z}) = \oplus_{x \in \Omega(X)} \phi(x\mathbf{z})$, for all $\mathbf{z} \in \Omega(\mathbf{Z})$. Clearly, solving #opt corresponds to computing: $(v^*, c^*) = \bigoplus_{X_1} \cdots \bigoplus_{X_n} \bigotimes_{\alpha \in F} \phi_\alpha$, where $v^*$ is the optimal solution cost and $c^*$ is the number of optimal solutions. This can be done using bucket elimination and search based algorithms such as those presented in the previous section.

**Mini-Bucket Approximation and #opt**  Mini-Bucket Elimination (MBE) is a classic relaxation of the exact bucket elimination that approximates each elimination operator to enable the user to control a bound on the space and time complexity [14]. The idea is to partition each bucket into smaller subsets called *mini-buckets*, each containing at most $i$ distinct variables (where $i$ is a user selected parameter called the $i$-bound). The mini-buckets are processed independently by the same variable elimination procedure resulting in messages over fewer variables and thus requiring less time and memory (namely, $O(n \cdot k^i)$, where $n$ is the number of variables and $k$ bounds the domain size).

The distributivity property from Proposition 1 allows us to extend MBE to the #opt task as well. Specifically, let $X_k$ be the current variable. The bucket $\mathcal{B}_k = \{\phi_1, \ldots, \phi_m\}$ is partitioned into $r$ mini-buckets $\mathcal{Q}_k = \{Q_{k1}, \ldots Q_{kr}\}$ such that $Q_{kj} = \{\phi_{j1}, \ldots \phi_{jl}\}$. Then, the exact elimination of $X_k$ from bucket $\mathcal{B}_k$, namely $\bigoplus_{X_p} \bigoplus_{\phi \in \mathcal{B}_k} \phi$ can be approximated by $\bigotimes_{Q_{kj} \in \mathcal{Q}_k} \bigoplus_{X_k} \bigotimes_{\phi \in Q_{kj}} \phi$. Unfortunately, this computation does not provide a bound on the number of optimal solutions. In fact, we can show that the number can go up or down in an unpredictable manner.

Consider for example the following valuations $a_1 = (2, 1)$, $b_1 = (1, 1)$, $a_2 = (2, 1)$ and $b_2 = (2, 1)$. We would like to provide a bound on the exact computation $(v^*, c^*) = (a_1 \otimes b_1) \oplus (a_2 \otimes b_2)$ by $(v, c) = (a_1 \oplus a_2) \otimes (b_1 \oplus b_2)$. In this case, we have that $(v^*, c^*) = (3, 1) \oplus (4, 1) = (3, 1)$ and $(v, c) = (2, 2) \otimes (1, 1) = (3, 2)$. Clearly, $c = 2$ is an upper bound on $c^* = 1$. On the other hand, if $a_1 = (2, 1)$, $b_1 = (1, 1)$, $a_2 = (1, 2)$ and $b_2 = (2, 1)$ then $(v^*, c^*) = (3, 1) \oplus (3, 2) = (3, 3)$ and $(v, c) = (2, 2) \otimes (1, 1) = (2, 2)$ in which case $c = 2$ is a lower bound on $c^* = 3$.

This observation is in stark contrast with what we know about other tasks for graphical models such as counting *all* solutions and finding the optimal solution to the model, where the MBE scheme is guaranteed to produce an upper and, respectively, a lower bound on the value of the exact computation. Therefore, we leave the extension of MBE into a correct bounding scheme for #opt as future work.

## 6 Experiments

We evaluate empirically our proposed counting algorithms on four benchmarks for graphical models. All experiments were run on a 2.6GHz processor with 10GB of RAM.

**Benchmarks and Algorithms**  For our purpose, we considered two random problem domains: (1) grid which consists of random $m$-by-$m$ grid networks, and (2) random which consists of random networks with $n$ variables and $2 \cdot n$ binary functions, respectively. We generated random problem instances for each domain, as follows: for grid problems, $m$ ranged between 8 and 14, respectively (so that the number of variables varied between 64 and 196); for random problems, the number of variables ranged between 60 and 120, respectively. In all cases, the domain size of the variables was set to 3. The function values were distributed uniformly at random between 1 and 10. In order

Table 1: Results for `grid` (left) and `random` (right) networks.

| | | grid instances | | | | | | | | | | random instances | | | | | | | | | |
|---|---|---|---|---|---|---|---|---|---|---|---|---|---|---|---|---|---|---|---|---|---|
| size | | p = 0.20 | | | p = 0.50 | | | p = 0.80 | | | size | | p = 0.20 | | | p = 0.50 | | | p = 0.80 | | |
| n, w* | alg | N | #opt | time | N | #opt | time | N | #opt | time | n, w* | alg | N | #opt | time | N | #opt | time | N | #opt | time |
| 64, 10 | AOBB | 10 | 13 | **0.08** | 10 | 267 | **0.07** | 10 | 5.58E+09 | **0.09** | 60, 11 | AOBB | 10 | 10 | **0.33** | 10 | 591 | **0.41** | 10 | 1.11E+07 | **0.35** |
| | A* | 10 | 13 | **0.08** | 10 | 267 | 0.09 | 0 | 0 | - | | A* | 10 | 10 | **0.33** | 10 | 591 | 0.48 | 9 | 9.27E+05 | 370.66 |
| | BE | 10 | 13 | 0.27 | 10 | 267 | 0.27 | 10 | 5.58E+09 | 0.27 | | BE | 10 | 10 | 5.85 | 10 | 591 | 14.91 | 10 | 1.11E+07 | 6.86 |
| | BnB | 10 | 13 | **0.08** | 10 | 267 | 0.09 | 6 | 2.25E+08 | 206.02 | | BnB | 10 | 10 | 0.34 | 10 | 591 | 0.49 | 10 | 1.11E+07 | 83.51 |
| 100, 13 | AOBB | 10 | 45 | **0.26** | 10 | 1,030 | **0.25** | 10 | 5.71E+12 | **0.26** | 80, 16 | AOBB | 10 | 56 | **1.72** | 10 | 1,201 | **1.40** | 10 | 1.16E+08 | **1.21** |
| | A* | 10 | 45 | 0.33 | 10 | 1,030 | 0.56 | 0 | 0 | - | | A* | 10 | 56 | 7.72 | 10 | 1,201 | 4.78 | 8 | 1.12E+07 | 1085.97 |
| | BE | 10 | 45 | 8.29 | 10 | 1,030 | 8.49 | 10 | 5.71E+12 | 9.92 | | BE | 4 | 7 | 2260.07 | 4 | 238 | 2213.98 | 3 | 2.38E+08 | 2617.60 |
| | BnB | 10 | 45 | 0.31 | 10 | 1,030 | 0.36 | 0 | 0 | - | | BnB | 10 | 56 | 1.91 | 10 | 1,201 | 2.18 | 9 | 7.98E+07 | 1021.19 |
| 196, 19 | AOBB | 10 | 182 | **199.32** | 10 | 9.71E+06 | **62.13** | 10 | 5.11E+22 | **113.34** | 120, 22 | AOBB | 7 | 320 | 1815.84 | 9 | 1.38E+05 | **1147.63** | 10 | 1.14E+13 | **182.24** |
| | A* | 2 | 96 | 2887.35 | 2 | 2,352 | 2885.78 | 0 | 0 | - | | A* | 0 | 0 | - | 0 | 0 | - | 1 | 1.59E+07 | 3320.61 |
| | BE | 0 | 0 | - | 0 | 0 | - | 0 | 0 | - | | BE | 0 | 0 | - | 0 | 0 | - | 0 | 0 | - |
| | BnB | 8 | 209 | 1025.42 | 7 | 1.20E+07 | 1635.57 | 0 | 0 | - | | BnB | 6 | 149 | 2029.35 | 7 | 1.63E+05 | 1598.08 | 0 | 0 | - |

Table 2: Number of optimal solutions (#opt) and CPU time (sec) on ISCAS (left) and SPOT5 (right) instances.

| | ISCAS instances | | | | | | | SPOT5 instances | | | | | | |
|---|---|---|---|---|---|---|---|---|---|---|---|---|---|---|
| instance | n, k, w* | #opt | AOBB | A* | BE | BnB | instance | n, k, w* | #opt | AOBB | A* | BE | BnB |
| c432 | (432, 2, 28) | 32 | **0.07** | 0.54 | - | 1050.67 | 1502 | (209, 4, 6) | 4.66E+70 | 0.08 | - | **0.03** | - |
| c499 | (499, 2, 24) | 32 | 39.60 | **17.10** | 118.03 | - | 29 | (82, 4, 14) | 1.13E+12 | **7.29** | - | 187.78 | - |
| c880 | (880, 2, 27) | 32 | **1.27** | 27.05 | - | 2783.92 | 404 | (100, 4, 19) | 21120 | **0.55** | - | 40.43 | 164.54 |
| s1196 | (1196, 2, 54) | 16 | **118.17** | - | - | - | 42 | (192, 4, 26) | 2.57E+11 | **2031.22** | - | - | - |
| s1423 | (1423, 2, 23) | 256 | **0.10** | 4.06 | 56.49 | 1843.46 | 503 | (143, 4, 9) | 6.41E+33 | **0.33** | - | 0.42 | - |
| s1488 | (1488, 2, 46) | 2 | 1.99 | **1.84** | - | 102.58 | 505 | (240, 4, 22) | 3.06E+42 | **3351.22** | - | - | - |
| s1494 | (1494, 2, 46) | 2 | **0.95** | 2.16 | - | 195.44 | 54 | (67, 4, 11) | 216 | **0.55** | 0.58 | 1.53 | 0.58 |
| s386 | (386, 2, 20) | 2 | **0.04** | 0.07 | 4.91 | 0.16 | | | | | | | |
| s953 | (953, 2, 72) | 32 | 99.67 | **5.50** | - | - | | | | | | | |

to control the number of optimal solutions we post-processed each function by randomly setting $p$ percent of the function values to 1. We refer to $p$ as the *perturbation* parameter and, intuitively, as $p$ increases, the number of optimal solutions should increase as well.

In addition, we also considered two collections of real-world WCSP instances derived from the ISCAS circuits [23] and SPOT5 satellite scheduling benchmark [24], respectively. For the SPOT5 instances the goal is to find the optimal schedule for an Earth observing satellite. Clearly, the number of optimal schedules may indicate certain degrees of freedom to operate the satellite in orbit. The ISCAS instances correspond to diagnosis of digital circuits where the goal is to compute the most likely explanation of a small subset of failed components. In this case as well the number of optimal solutions could explain the reliability of the circuits. The original problem instances which we obtained from the UCI Graphical Models Repository (`graphmod.ics.uci.edu`) are specified as Markov networks with real valued potential values between 0 and 1. We converted these instances into equivalent WCSPs by taking the negative log of the potential values and rounding up to the nearest integer value.

We evaluated algorithms BE and AOBB and compared them with the A* search and the depth-first branch and bound (BnB) that enumerate explicitly the optimal solutions. All search algorithms were by guided by a static mini-bucket heuristic MBE($i$) which was pre-compiled along a min-fill elimination ordering [14, 19, 25]. The heuristic uses a parameter called $i$-bound to control the accuracy of the heuristic estimates. We set the $i$-bound to 10 and allowed a 1 hour time limit to all algorithms.

**Measures of Performance** We report the average CPU time in seconds (time), the number of problem instances solved (N) and the average number of optimal solutions over the solved instances (#opt), respectively. In addition, we also collect the problems' parameters as the number of variables ($n$), maximum domain size ($k$), and the average induced width ($w^*$) obtained along a min-fill based elimination ordering [26]. The best performance points are highlighted. A "-" denotes that the respective algorithm exceeded the time or the memory limit.

**Results** Table 1 summarizes the results obtained on the `grid` and `random` domains. The left-most column indicates the problem sizes, while the remaining columns are divided into 3 groups corresponding to different values of the perturbation parameter $p \in \{0.20, 0.50, 0.80\}$. Each data point represents an average over 10 random instances. We can see that AOBB is the overall best performing algorithm both in terms of running time and the number of problem instances solved (over 96% of instances solved). Furthermore, algorithms A* and BnB are competitive only for problems

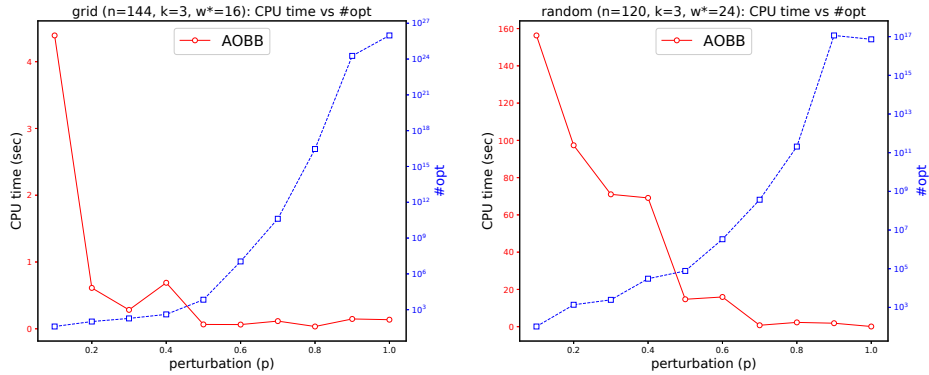

Figure 3: CPU time in seconds (red) versus number of optimal solutions (blue).

with small and moderate numbers of optimal solutions (e.g., grid and random with $p = 0.20$ and $p = 0.50$, respectively). However, they typically fail to solve most of the problem instances where the number of optimal solutions is very large (e.g., grid and random with $p = 0.80$) because of the prohibitively large overhead associated with enumerating the optimal solutions. Finally, BE is competitive only on problem instances with relatively small induced widths.

In Table 2 we show the results for solving the ISCAS and SPOT5 problem instances. We see again that AOBB offers the best performance, especially on the SPOT5 instances which have the largest number of optimal solutions. As before, BE can only handle problems with relatively small induced width regardless of the number of optimal solutions. In this case, the performance of A* and BnB is quite poor compared with AOBB and BE because of the large number of optimal solutions.

Figure 3 plots the running time of AOBB and the number of optimal solutions as a function of the perturbation value ($p$) for two representative problem classes from the grid and random domains. Each data point represents an average over 100 random instances generated for the respective $p$ value. We see that as the number of optimal solutions increases (i.e., $p$ increases), the problems become easier to solve and the running time decreases. This is important especially when designing random problem generators for optimization to control the hardness of the problem instances generated.

## 7   Related Work

Model counting (#SAT), solution counting (#CSP) or weighted model counting (WMC) are well known #P-complete problems that have many applications in fields such as verification, planning and automated reasoning. Exact approaches to counting solutions are based on either extending systematic search-based SAT/CSP solvers such as DPLL and AND/OR search [27, 28, 29, 30], or variable elimination algorithms [14] which are known to be time and space exponential in the induced width of the problem. Our algorithms build on top of those ideas by combining summation and optimization without resorting to explicit enumeration. Approximate model counting techniques based on hashing were recently proposed [31, 32]. However, these methods are not directly applicable to the #opt task, at least not in a straightforward manner. Maximum model counting (Max#SAT) is a recent extension of #SAT [33] that is also related to #opt. The work by [34] develops a semiring based formalism for counting weighted subgraphs in an explicit larger graph.

## 8   Conclusion

We introduced here the #opt task for graphical models, presented and evaluated variable elimination and depth-first AND/OR branch and bound algorithms for this task. We also described a semiring based formulation of the task. The complexity of the proposed algorithms is exponential in the induced width and does not depend on the number of optimal solutions. Our empirical evaluation demonstrated their effectiveness compared with brute-force search approaches that rely on explicitly enumerating the optimal solutions. Overall, our proposed AOBB version appears to be superior.

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
