[Reviews · NeurIPS 2019]

Reviewer 1



The paper deals with the problem of counting the number of optimal solutions in weighted CSPs. In order to solve it without requiring to enumerate these solutions, they map the cost functions into pairs (cost,count) and therefore reformulate the optimization problem within Semiring (R^2,x,+). They show how to adapt bucket elimination and AND/OR Branch and Bound to this semiring. Finally, some experiments highlight the effectiveness of the method. Although optimization in semirings is not novel and the formulation provided by the authors is not very surprising, the methodology is interesting by itself. The experiments are convincing on the set of algorithms compared. However, there exist in the literature other methods whose purpose is specifically to count models, e.g., the weighted model counting algorithm used by Chavira and Darwiche, but also sum-product networks in which MPE-like queries can be computed efficiently. Why don't the authors compare their algorithms against them in the experimental section?

Reviewer 2



The authors deal with a new problem that extends existing problems in a natural way, and present new solutions that extend existing solutions in a natural way; thus in some ways the paper is not very original, but it must be noted that the problem #opt has some surprising properties that make the whole endeavor quite interesting. As for the contribution, it is a valuable piece with new algorithms that have good performance. I really take issue with the emphasis on semirings; I cannot see how this helps the whole effort. Are semirings bringing new insights here? Or are they just more general so that the algorithm applies to other problems? Why does it make a difference to state results this way? Concerning the significance, the results do address and solve a new problem, but I think the authors should perhaps work a bit harder in explaining why this problem is significant in practice. This is a point that is left a bit weak. Concerning the text: - Instead of "[6] introduced...", better "Ref. [6] introduced..." (and similar expressions). - The fonts for Algorithm 1/2 are **very** small... please increase them. - Also Figure 3 is really too small. - Page 7, line 300: "were [by] guided by". - References: "fortran" should be "Fortran".

Reviewer 3



I have not much to say about this work for it is really good. The only sad thing is, that the authors ignore some classic results on this topic, like: "Finding the M Most Probable Configurations Using Loopy Belief Propagation", Chen Yanover and Yair Weiss, NIPS 2003

[Author Response · NeurIPS 2019]

**Review #1:** Thanks for your comments and suggestions.

We are aware of the weighted model counters by Chavira and Darwiche which are based on either search or knowledge compilation ideas. However, these algorithms count **\*all\*** solutions whereas our proposed algorithms focus on counting **the optimal solutions** only. It is important to emphasize that counting optimal solutions in a compiled structure such as a decision diagram (e.g., arithmetic circuit, multi-valued AND/OR decision diagram, OBDD) require two passes over the compiled structure and therefore will yield the count of optimal solutions by enumeration. In contrast, our algorithms are not dependent on the number of optimal solutions and therefore are much more efficient as we demonstrate in the empirical section.

Regarding the sum-product networks, we believe that they fall within the same category of algorithms that count all solutions. In principle, we think that they can also be extended to count optimal solutions but they will compute the number of optimal solutions by enumeration as well.

Finally, extending the current weighted model counters and sum-product networks to counting optimal solutions is not yet clear and therefore will be considered as part of our future work.

**Review #2:** Thanks for your comments and suggestions.

The semiring is a general framework that allows us to specify a wide range of reasoning tasks for deterministic and/or probabilistic graphical models in a unified manner. In this paper we show for the first time that counting the number of optimal solutions can be formulated within this general framework as well and solved exactly with variable elimination and search algorithms. Therefore, our algorithms can be extended to more general semiring based graphical models. For example to finding the number of optimal policies in an influence diagram.

We also show that, unlike many other reasoning tasks that are formulated within the semiring framework and admit simple partitioning based bounding schemes (e.g., using for instance the mini-bucket approach), the #opt task cannot be approximated using these kinds of schemes to produces bounds (upper or lower) on the number of optimal solutions.

We will fix all typos and presentation issues. Regarding the small fonts used for the algorithms, we will use an extra page to avoid all the space issues. This will also allow us to expand on the motivation behind our work and discuss additional examples where we believe that #opt is important.

**Review #3:** Thanks for your comments and suggestions.

We are aware of the Yanover and Weiss work on finding the M most probable configurations and we will definitely cite it. In principle, this work together with the more recent work by Flerova, Marinescu and Dechter (which we cite) on finding the M best solutions using search can be extended to compute the number of optimal solutions but these specialized m-best algorithms will compute the count of optimal solutions by enumeration which is much less effective (especially on problems with many optimal solutions) than our proposed algorithms which do not depend on the actual number of optimal solutions.

In the extreme case where there is only 1 optimal solution a search based m-best algorithm should be as good as our AOBB approach for #opt. In all other cases, the m-best algorithms must iterate for several values of m in order to find the number of optimal solutions. Therefore, we emphasize that our AND/OR branch and bound for #opt is always better than a specialized m-best approach. We will extend the discussion in the paper to clarify this point.

[Meta-Review · NeurIPS 2019]

A solid contribution through a smart formulation to count optimal MAP solutions in graphical models while avoiding to enumerate them. Feedback is satisfactory, even though there were no substantial open questions that would likely change the faith of the manuscript. Techniques are somehow well-known, but the problem is interesting and their application there required some clever work.